# Magnetic Railway Sleeper Detector

Lukas Heindler [1,*], Harald Hüttmayr [2], Thomas Thurner [1] and Bernhard Zagar [3]

1 Chair of Automation and Measurement, Montanuniversität Leoben, 8700 Leoben, Austria; thomas.thurner@unileoben.ac.at
2 System7 Railtechnology GmbH, 4664 Laakirchen, Austria; harald.huettmayr@s7-rail.com
3 Chair of Electrical Engineering, Montanuniversität Leoben, 8700 Leoben, Austria; bernhard.zagar@unileoben.ac.at
* Correspondence: lukas.heindler@stud.unileoben.ac.at

**Abstract:** In an ever expanding railway network all around the world, the need for track maintenance grows steadily. Traditionally, one major part of track maintenance is ramming large vibrating steel picks into the gravel between and under railway sleepers to compress the gravel and generate a safe substructure. Even today, maintenance personnel still have to manually locate the sleepers if they cannot be detected by computer vision systems or visually by the operator. Here we developed a first of its kind magnetic sleeper detector, even able to find sleepers, buried in gravel, undetectable by vision based systems. Our approach of magnetic detection is based on a DC magnetic field excitation and a detector moving with respect to the rail system, including the sleepers and fasteners for mounting the rails. Due to railway application constraints a large air gap between the sensor and the sleeper structure is required, which significantly complicates the magnetic sensing task for robust sleeper detection. The design and optimization of the magnetic circuit was based on extensive 3D simulation studies to ensure highest possible variation in magnetic flux density at the sensor locations for absence and presence of a sleeper. Furthermore, a low noise and high sensitivity electronic circuit has been realized to cope with sensor signal offsets from unknown or changing sensor orientations with respect to the earth's magnetic field, or magnetic interferences from other trains potentially passing by during active measurements. Since we only want to detect sleepers in close vicinity of the moving sensor system, digital signal processing of the acquired signals can easily compensate for disturbing slowly changing or static field components within real world application scenarios. We demonstrate that magnetic detection of even buried sleepers on railway tracks is possible for distances of up to 172 mm between the sensor and the sleeper. This enables an even higher level of railway maintenance automation previously impossible in certain scenarios.

**Keywords:** magnetic sensing; railway maintenance; Hall effect sensor; magnetic anomaly detection

## 1. Introduction

As the need for environmentally friendly cargo and passenger transportation steadily increases, railway systems are currently built or expanded all over the world. With growing railway networks, more and more tracks have to be maintained and serviced. One major part of track maintenance is ballast compaction. Ballast compaction describes the action of ramming large metal picks into the freshly distributed gravel beneath, around and above the sleepers, on which the tracks are placed. By vibrating the picks until the gravel is adequately compressed, a safe subfloor is constructed for continued traffic. Such picks are driven by tamping units, machines/trains like the one seen in Figure 1. Tamping machines feature tamping units that drive several, in this case, picks into the gravel while vibrating the picks for better gravel distribution. With vibration frequencies around 35 Hz the gravel acts viscous, guaranteeing an optimal tamping result.

Most of the time the sleepers are exposed, not covered by the gravel and so detectable by the tamping crew. From time to time however, it happens that the gravel covers the

sleepers entirely, undetectable by any vision based systems. In such cases, maintenance personnel has to set the tamping machine by hand and estimate next allowable picks' positions—in between sleepers—from previous picks' positions. This task is more time consuming and in need of skilled personnel. Additionally, higher costs due to longer lasting maintenance work and destruction potential of sleepers or other parts of the track by inadvertently ramming the picks into them can constitute a considerable amount of financial risk of the procedure.

Hence, a new, more general approach to sleeper detection is needed. A unit not relying on visual cues could enable automation of sleeper detection even if the whole track was covered by gravel. It would free the operator from an inspection task thus potentially increasing operational speed and still reducing the number of man hours per kilometer processed just needed to find sleeper positions. It would also increase occupational safety during maintenance. Additionally, and besides building a new railroad line, the maintenance of existing lines could be accelerated leading to reduced maintenance costs. In addition, repair work of for whatever reason destroyed sleepers could drastically be reduced. Furthermore, an increased automation of the tamping process can counter personnel shortages that start to limit new projects already.

Nowadays, the state-of-the-art procedure of sleeper detection, besides direct visual detection by the tamping crew, uses LiDAR units. Such units scan the railway tracks and deliver with resolutions down into the mm–range three dimensional mappings of the floor plan consisting of the gravel, the sleepers and the rails, including the rail fasteners. By processing these maps, sleepers can be detected reliably in those situations where sleepers are exposed. In certain scenarios where there is gravel not only beneath the sleeper but also covering them, this technique delivers no useful data, however. For such cases, there has been little progress in the railway maintenance business.

Ground penetrating Radar [1] is a further possibility to detect hidden structures, in our case the gravel covered sleepers. Since there is a great variety of sleeper materials (oak sleepers, pre-stressed concrete sleepers, steel sleepers, plastic sleepers, etc.) the performance of ground penetrating radar is presumed to be rather limited.

In contrast to previously mentioned systems, we are not aiming at the detection of the sleepers per se but at the practically always identical ferromagnetic sleeper screws using a sort of magnet camera [2]. This approach has the advantage of being able to locate sleepers with a longitudinal precision in the range of 10 mm to 30 mm independently of the amount of gravel covering the sleepers since the screws have small diameters and their axes locations are rather easy to be estimated accurately.

The major problem encountered in the magnetic way of finding covered sleeper screws is the railroad clearance profile that must be adhered to under all circumstances. This leads to air gaps in the to-be-designed magnetic circuit above 150 mm and resulting magnetic flux densities not much exceeding the earth's magnetic field or the induced magnetic AC fields caused by the traction currents of passing trains. We address these problems by designing a sensor system able to cope with these circumstances using magnetic field concentrators, appropriately selected magnetic field sensors, and a particular signal processing scheme that is adapted to air gap width fluctuations.

The paper is structured as follows: in Section 2 the design process of the detector based on [3] is elaborated on, including the magnetic circuit design, the electrical circuit design, the simulation results of the magnetic circuit, as well as the mechanical and structural design of the case. Section 3 discusses the sensor characterization, the measurement set-up, the different measurement scenarios, and several adaptions of the set-up circumventing experienced problems of the prototypes. The measurement results are reviewed in Section 4. Finally, in Section 5 results and possible improvements are discussed and the paper is concluded.

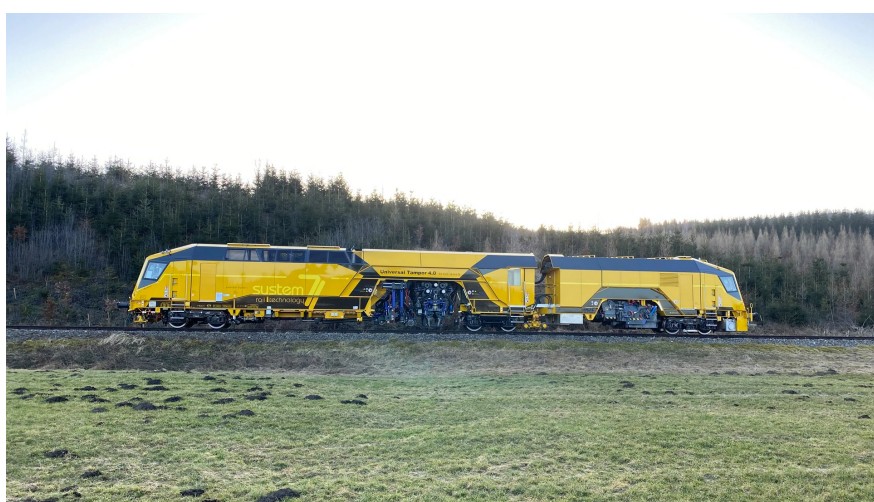

**Figure 1.** Tamping machine from the company system7 [4]. It features 16 picks which are continuously driven into the gravel between the sleepers to compress it and build a solid track foundation.

## 2. Detector Concept and Magnetic Circuit Design

Visual detection of sleepers is rather impossible in certain scenarios as outlined in the previous section, thus a method of detection better adapted to railway requirements has to be implemented. Sleepers nowadays are made out of several materials like wood, steel, plastics, concrete, and today mostly steel–reinforced concrete. Therefore, a single detection method to identify the sleepers themselves seems hardly possible. However, as indicated above, the fastenings of the rails to the sleepers, as exemplarily shown in Figure 2, is besides its shaping and specific configuration, always made of the same kind of structural steel [5]. Therefore, we elaborate on a novel sensor system aimed at detecting the shape of the fastener and/or the fastening screw itself which is protruding out of the fastening assembly.

Taking into account that steel can exhibit a certain magnetic remanent field, an AC excitation might be present from passing trains, and the railroad clearance profile prevents close proximity of the magnetic field sensor and the fastener. The layout of the magnetic circuit turned out to be rather involved, since it must accommodate all external fields (earth's magnetic field in arbitrary direction, strong AC fields, and remanence fields of pre-magnetized components in the vicinity of the rail structure) and still operate reliably while maintaining a sufficiently high signal–to–noise ratio (SNR) and high robustness against external magnetic interference. For these difficult cases metrological tricks are available by transforming the useful signal in the frequency domain way off any interfering signal's frequency band by employing the carrier frequency method [6]. Utilizing this method in the given application was rather difficult since most construction materials in railway tracks are electrically conductive and any applied AC magnetic field induces immense eddy currents, dissipating most of the introduced magnetic power and still giving negligible measurement signals. Thus, our approach was chosen to be based on static (DC) magnetic excitation and a moving sensor system mounted on track building machines operating with velocities in the range of $5 \, \mathrm{km \, h^{-1}}$, which would only cause sufficiently small eddy currents in metallic rail track components.

The required air gap in the range of $150 \, \mathrm{mm}$ and more excludes the potential use of an electrically excited magnet due to the huge required excitation in the order of several hundred thousand ampere windings. Therefore, we integrated a suitable permanent magnet into our sensing magnetic circuit.

As a consequence of the clearance profile, which sets the boundary for any obstacle in the train's way, and scenarios in which the newly distributed gravel can reach up to the top surface of the railway track, the lowest part of the detector has to be above the rail's top surface. This fact leads, as already indicated, to an air gap between the magnetic field detector of the sensor and the fastening bolts of up to $170 \, \mathrm{mm}$. As a result of this

huge required air gap, particularly for a magnetic detection method, optimizations for best performance are necessary. Due to the complexity, time and resource consumption of magnetic simulation studies as the basis for any magnetic circuit optimization, we reduced the investigated configurations to a limited number of variations of only the most promising geometrical parameters of the setup.

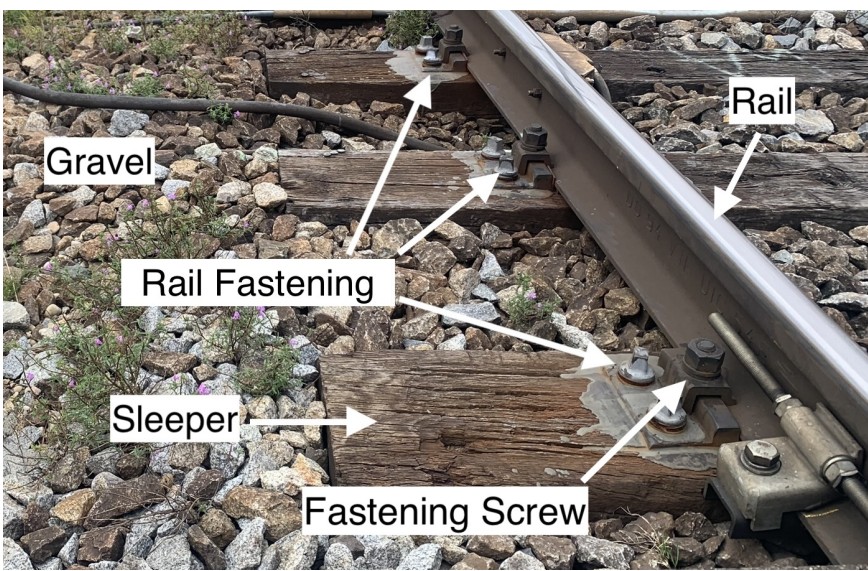

**Figure 2.** Rail Fastening in this case clamping rails to sleepers made of wood.

This section elaborates the magnetic, electrical and mechanical design processes. Beginning with the magnetic circuit design, a proposed magnetic circuit is optimized via simulation runs and variations of the free-to-choose parameters magnet length $\ell_m$ and arm length $\ell$ (see Figure 3). Based on the expected magnetic flux density variation and its resulting sensor output voltage swing, an adaptive electronic circuit with high-precision analog signal processing is designed. Around the magnetic and electrical circuits the mechanical structure, including the detector case and the screwing, is carefully developed. Hall effect sensors are used to measure the perpendicular magnetic flux density field lines.

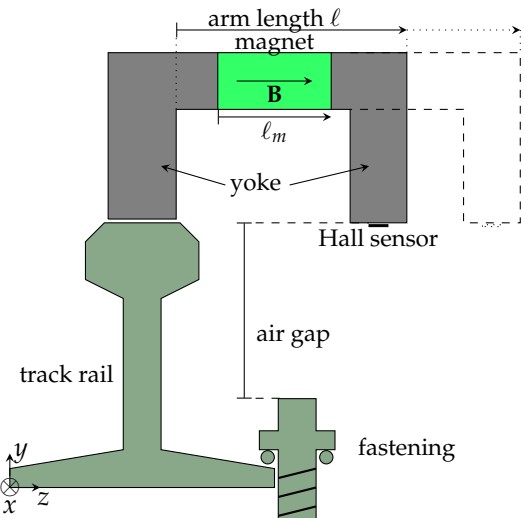

**Figure 3.** Along the track view of the finally chosen magnetic circuit. In green the permanent magnet, in gray the yoke of the magnetic circuit, whose geometry has to be optimized, and in emerald green all the possible ferromagnetic structures of the railroad track.

### 2.1. Magnetic Circuit Design

There are many possible ways to detect steel parts in close proximity. For greater distances, the number of possibilities decreases. Therefore, we chose a DC magnetic detection approach which in turn measures the magnetic flux density variation due to the absence or presence of the steel fastening under the detector when it is moving by on the tamping machine. We propose a magnetic detector consisting of a magnetic circuit including a magnetic excitation by a permanent magnet, a substantial air gap due to the railway clearance profile, the rail and the fastenings, as well as a magnetic field sensor as shown in Figure 3. Figure 4 shows the front view of the detector with the detector moving by the fastening in rail direction. This magnetic circuit is proposed to be optimized for highest possible magnetic flux density variation as recognized by the sensors *sen*1 to *sen*3 while not collecting metal parts from the railway tracks.

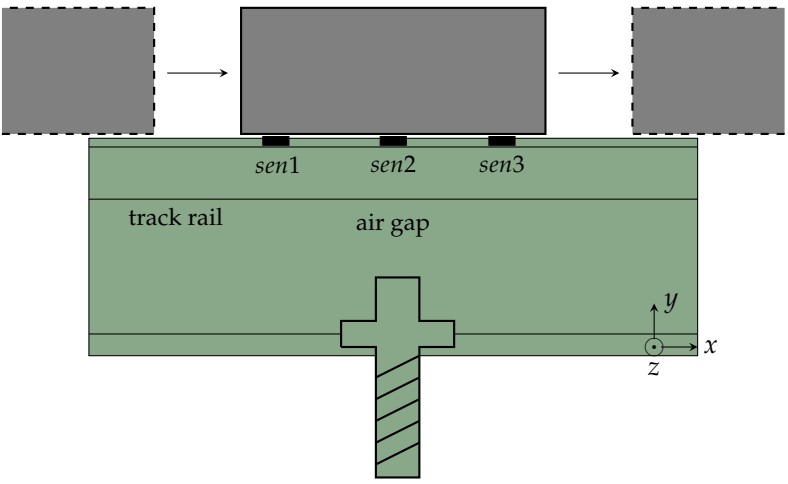

**Figure 4.** Frontal view of the magnetic circuit with the sensor assembly equipped with three magnetic sensors (Hall effect sensors) indicated by *sen*1, *sen*2 and *sen*3 passing by the fastening.

The design of a magnetic circuit is especially challenging as a result of the highly non-linear behavior of the ferromagnetic materials in the magnetic circuit. Furthermore, a significant air gap, inevitable due to the clearance profile, introduces another major uncertainty in the distribution of the magnetic flux density of the detector. Therefore, three-dimensional finite element method (FEM) simulations were carried out using the FEM tool *Maxwell 3D* (Ansys, Inc., Canonsburg, PA, USA) [7] with a simple magnetic circuit as shown in previous Figures. The novelty of this magnetic detector is not specifically its geometry, but its tailoring to the given application needs. As constrained by the railway clearance profile, especially the magnetic circuit's simple structure should enable fastening detection even at significant distances. More complex geometries could potentially increase detection probability, but far more complex simulations would have to be carried out. As shown in Figure 5, the model of the track, the detector and the fastening were modeled within the simulation environment. The model parameters are given in Table 1. The magnet, indicated in light green, acts as the magnetic excitation. The goal of the simulation was to explore the field's behavior when the fastening is moving by under the detector, simulating a tamping machine going over sleepers. The simulation study was carried out as a sequence of single field simulations for scenarios from step-wise changes of the distance between the sensor and the fastening bolt, with steps of 10 mm in $x$-direction from $x = -200$ mm to $x = 200$ mm (see Figure 4). At each step the field was then simulated. Also, the model was equipped with three magnetic sensors at crucial positions on the yoke's air gap facing side as sketched in Figure 4. These positions are specifically chosen to evaluate the magnetic flux distribution inside the yoke. The sensors at the edge of the yoke have

been added to assess edge effects of the magnetic field, whereas the central sensor provides information about the magnetic flux density in the center of the joke. Comparing magnetic flux densities at the three sensor positions allows the evaluation of magnetic flux density differences for specific application situations, which is important for the sensor placement in the realized prototypes. More sensors would drastically increase simulation complexity without delivering more relevant information for the final sensor design. Besides, the simulated sensors have been located for sensing the field at three positions, evaluated only in *z*-direction (concerning the coordinate system defined in Figure 3). The strongest field variation due to the fastening was measured at the inner edge of the yoke. As a result, only those three positions were then used for further simulations. It should be noted that using two sensors, as in the prototype, is the minimum requirement for this detection application. Having three or more sensors would allow for the measurement of additional railway properties such as additional steel lying around or the position of the detector with respect to the fastening (before or after). Moreover, if even more sensors were used, possibly spread over a distance equivalent to the standard sleeper distance, it would be possible to measure the actual sleeper distance and potentially estimate the position of the next sleeper. Two sensors were chosen due to constraints on detector size and cost.

The obtained simulation results, as shown exemplarily in Figures 6 and 7, reflect the expected behavior. In Figure 6 the magnetic flux density field for $x = 0$ mm is displayed as provided by the simulation study. Only minor field strengths are observed at the fastening which leads to the assumption of a small signal variation at the position of the sensors. In Figure 7 the magnetic flux at the three sensor positions are visualized while having the relative motion of the rail (thus the sleeper fastening) with respect to the sensor. For sensor 1 the highest magnetic flux is measured at $x \approx -80$ mm before the other sensors, sensor 2 and 3, reach their maximum at $x = 0$ mm and $x \approx 80$ mm, respectively. As can be seen, the flux difference between a present and an absent fastening of $\Delta\Phi \approx 3$ nWb is as expected only minimal. Converting 3.2 nWb to a magnetic flux density difference yields

$$\Delta B = \frac{\mathrm{d}\Delta\Phi}{\mathrm{d}A} \approx \frac{\Delta\Phi}{A} = \frac{89.8\,\mathrm{nV\,s} - 86.6\,\mathrm{nV\,s}}{4\,\mathrm{mm}^2} = 0.8\,\mathrm{mT}\ , \tag{1}$$

where $\Delta B$ is the magnetic flux density and $A$ the simulated sensor area of $4\,\mathrm{mm}^2$. As a result, a magnetic sensor would require a resolution much better than 0.8 mT with a useful measurement range of at least 91 nWb, which corresponds to approximately 23 mT in this configuration for a simulated magnet exhibiting a zero air gap flux density of 1 T.

The obtained magnetic field changes, which are to be sensed by the used magnetic Hall effect sensors, are quite small with respect to the static ambient fields and disturbances, thus the sensor electronics requires for an optimized low noise electronic circuit design to enable robust sensing of magnetic flux density differences $\Delta\Phi$ from the magnetic sensor elements.

**Table 1.** Material parameters as used in the simulations.

| Parameter | Value |
|---|---|
| Magnet kind | N48H |
| Magnet remanence $B_r$ | 1.4 T |
| Magnet coercive force | 955,000 A/m |
| Material detector | 1010 steel [5] |
| Material rail | 1010 steel [5] |
| Material fastening | 1010 steel [5] |

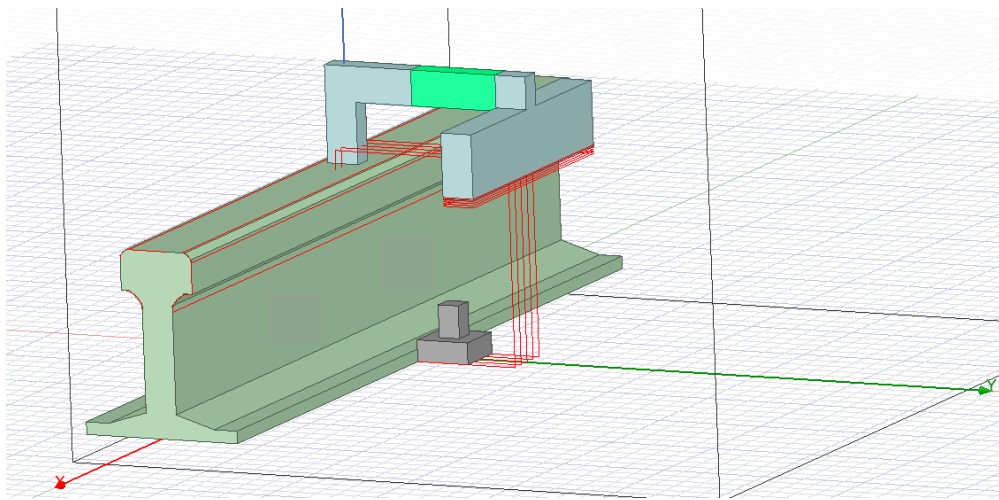

**Figure 5.** Model for the simulation of the magnetic flux density distribution of the proposed magnetic circuit of the sleeper detector. Red lines delimit areas of high interest that are discretized much finer to yield more accurate flux density simulations.

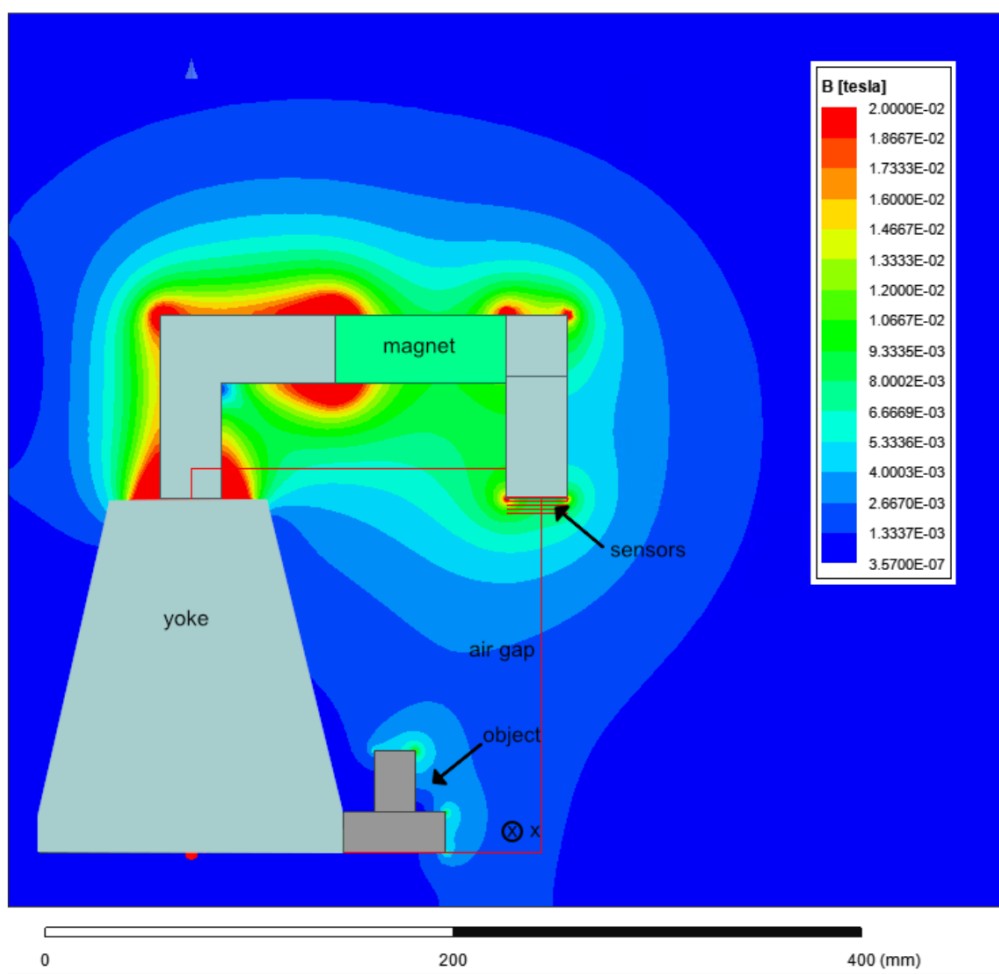

**Figure 6.** Magnetic flux density field simulated for a fastening position $x = 0\,\text{mm}$. Only minor fields are present at the fastening, hinting at low field strength differences at the sensor positions. Red lines delimit areas of high interest that are discretized much finer to yield more accurate flux density simulations.

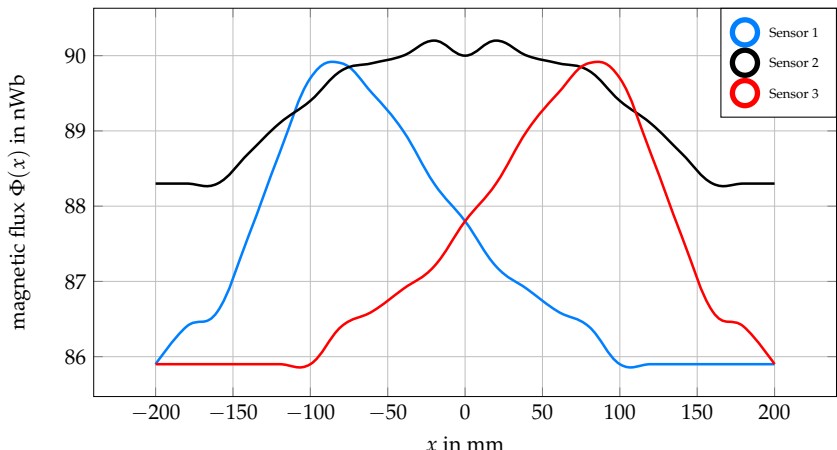

**Figure 7.** Simulation of the magnetic flux density at the positions of sensor 1 to 3 when the fastening is moved by under the detector. At $x \approx -80\,\text{mm}$ sensor 1 reaches its maximum before sensor 2 and 3 reach theirs at $x = 0\,\text{mm}$ and $x \approx 80\,\text{mm}$ respectively. The higher pedestal, or offset from sensor 2 can be explained by a higher flux density at the center of the detector.

The optimization problem in this scenario is the large dimensionality of the optimization problem. Without any parameter being especially obvious to vary to approach the global optimum, and limited computer resources, simulation capacities are limited. Therefore, a single parameter (the arm length parameter $\ell$ in Figure 3) was chosen to be varied in the optimization process. Keeping in mind that the magnetic field will minimize its respective total energy content depending on the positioning of ferromagnetic elements, the magnetic circuit has to be optimized in a way that the fastening is interspersed with as many field lines as possible. The simulation was set for arm lengths between $\ell = 130\,\text{mm}$ and $330\,\text{mm}$ in steps of $40\,\text{mm}$. The simulation arm lengths were chosen to be shorter than the outer edge of the track's clearance profile and as long as to reach over the fastening. Again the fastening was moved in $x$ direction from one limit to the other, for each arm length. These simulations resulted in curves shown in Figure 8. Plotted are six $\Delta\Phi_i(x) = \Phi_{i,\text{sen1}}(x) - \Phi_{i,\text{sen3}}(x)$ graphs, each with a different arm length $\ell$. With the highest $\Delta\Phi$ an arm length of $\ell = 170\,\text{mm}$ was chosen for future simulations and prototypes.

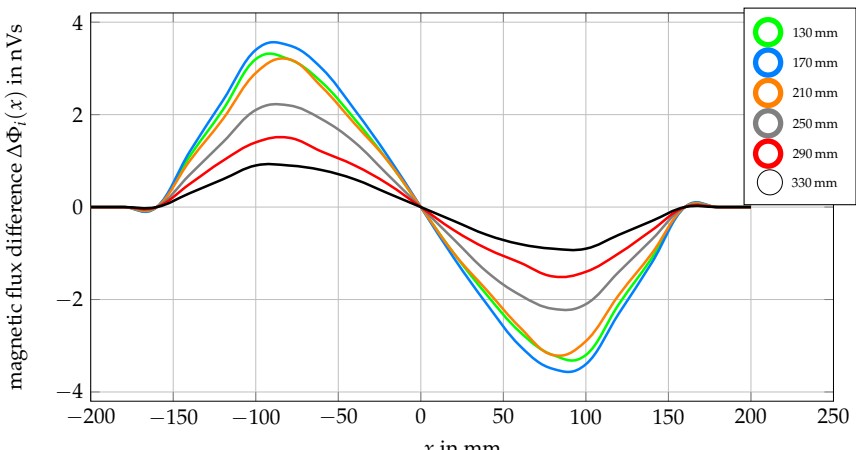

**Figure 8.** Difference $\Delta\Phi$ of sensor signals 1 and 3 when moving the fastening by under the detector with different arm lengths $\ell$. With higher $\Delta\Phi$ the fastening is easier to detect. For $\ell = 170\,\text{mm}$ the largest difference between fastening absent and present can be measured.

Besides, due to the fact that anything ferromagnetic in close proximity to the magnetic circuit can and will change the magnetic field's behaviour, even the mechanical housing of the detector has to be made from non-magnetic materials like aluminum. Especially the

yoke case and its screws have to be non-magnetic. Otherwise, unpredictable effects can occur which deteriorate the detection performance.

*2.2. Electronic Circuit Design*

The electronic design is based on the simulation results and the expected magnetic flux density variation at the sensor positions and realized in KiCAD [8]. Due to expected minute signal variations and the resultant output voltage, a rather large gain on the sensor signals has to be applied. Compensation methods are typically the way to go, here the sensor outputs of a leading and a trailing sensor compensating their respective pedestals. Therefore, only two, instead of the three sensors like in simulations are used with their output signals subtracted by one another by an instrumentation amplifier providing a high common mode rejection, thus allowing for a large gain on the differential mode signals. The two sensors are mounted in a way that the center of the yoke is at the center of the two sensors for equal flux density penetration. Three sensors were used in the simulations to get a feeling for the field distribution in and around the yoke.

Since the simulation results showed magnetic field lines perpendicular to the yoke's air gap facing side, the sensors had to be sensitive in direction perpendicular to the mounting surface. Additionally, the simulation showed very small changes of field variation between absence and presence of the fastening, which in turn demands high sensitivity and a specific measurement range. Due to their high sensitivity, range and excellent linearity, Hall sensors have been chosen as utilized sensor technology. Therefore, for the first detector prototype the Honeywell SS94A1F through hole technology (THT) Hall effect sensor [9] was used primarily for its high sensitivity of $250\,\mathrm{V\,T^{-1}}$. In this revision, however, the goal was to ease fabrication and minimize build size, which was accomplished by using surface mount devices (SMD). Hence, an SMD Hall effect sensor with similar properties was analyzed and tested to replace the THT Hall effect sensor. Due to its high sensitivity compared to other SMD Hall effect sensors and its analog output, the SiLabs Si7211 SMD Hall effect sensor [10] was chosen for comparison to the Honeywell sensor.

The Honeywell sensor delivers a ratiometric analog output centered between its supply voltage that can be chosen to lie between 6.6 V to 12.6 V, and clamped to the positive port. Additionally, it has a step response time (from 10 % to 90 %) of $3\,\mu s$ and a linearity of 1.5 % at max. in its linear operating range of $\pm 10\,\mathrm{mT}$. Due to the expected static magnetic field at the sensor positions of more than the sensor's range, a different magnet with a weaker effective magnetic flux density of 0.607 T was chosen for the first prototypes. The SiLabs sensor offers digital as well as analog outputs in an SOT-23 package. With a supply range of 1.7 V to 5.5 V the supply circuit has to be adopted. The sensor's sensitivity is half that of the Honeywell sensor at $125\,\mathrm{V\,T^{-1}}$.

Based on the sensors's electrical specification, the electronic circuit was designed as seen in Figure 9. The electronics consist primarily of the following blocks:

(a) Supply block: Traco 4822 DC/DC converter [11], which converts the supplied 24 V to $\pm 12\,\mathrm{V}$ to power the other electronics. Analog Devices REF01/02 voltage reference IC [12] combined with operational amplifiers [13] and bipolar junction transistors in a self regulating loop to supply the Hall effect sensors with a stable voltage,

(b) signal processing block: Analog Devices AD8421 instrumentation amplifier [14] with variable offset and single resistor programmable gain to compensate the pedestal signals of pairs of the of two Hall effect sensors and only amplify their difference.

Additionally, several bypassing capacitors, mounting holes, jumpers, connectors, resistors and other parts are added for proper functionality.

The working principle is as follows: The voltage references control the voltage for the voltage followers to within tight limits which themselves supply the sensors with high precision operating supplies, which the references are not designed to provide directly. The sensors's output signals are filtered with a first-order low-pass filter with a cut-off frequency of 1592 Hz to allow for higher detection speeds and effectively suppress AC traction current's interferences before the pedestal compensation is done by the instrumentation

amplifier. Due to the amplification of only the difference of the signals, a near zero voltage level at zero deflection is striven for. Therefore, an offset correction circuit is implemented which takes an external voltage signal through a voltage divider and a voltage follower, before it is passed to the instrumentation amplifier. With this method, offset can be adapted for each detector individually improving detection probability due to higher possible gain. Additionally, the offset pin can be grounded with a jumper in case no offset correction signal is available. The output signal of the electronic circuit is an analog amplified voltage signal ($[-10\,\text{V}\,;\,10\,\text{V}]$) to be processed by an external programmable logic device (PLD) with exactly that input voltage range. This PLD is responsible for the surveillance and processing of all sensor data of the tamping machine.

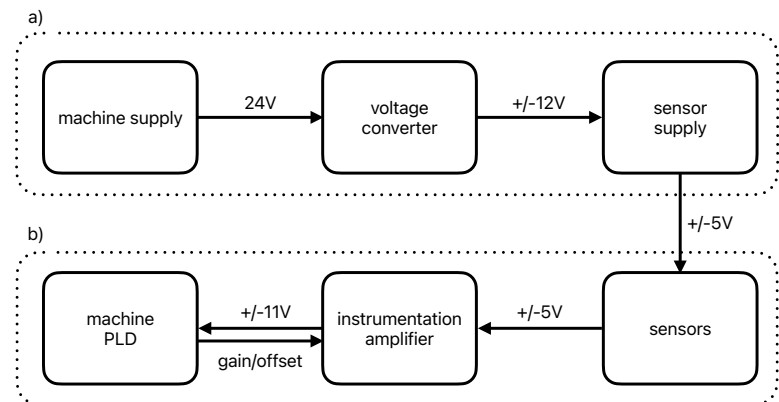

**Figure 9.** Block diagram of the current detector version featuring (**a**) the supply block and (**b**) the signal processing block.

Figure 10 shows the printed circuit board already installed in the current detector prototype. Attention was paid in particular on a compact design to ensure short routes for minimal parasitic effects and best performance. The grey cables lead down to the sensor board and supply the sensors, as well as return the sensors' output signals. The multicolored cables lead to the plug connection, seen on the left, which connects the board supply and the amplifier output to the supply cable. The board has a length of 90 mm and a width of 33.5 mm.

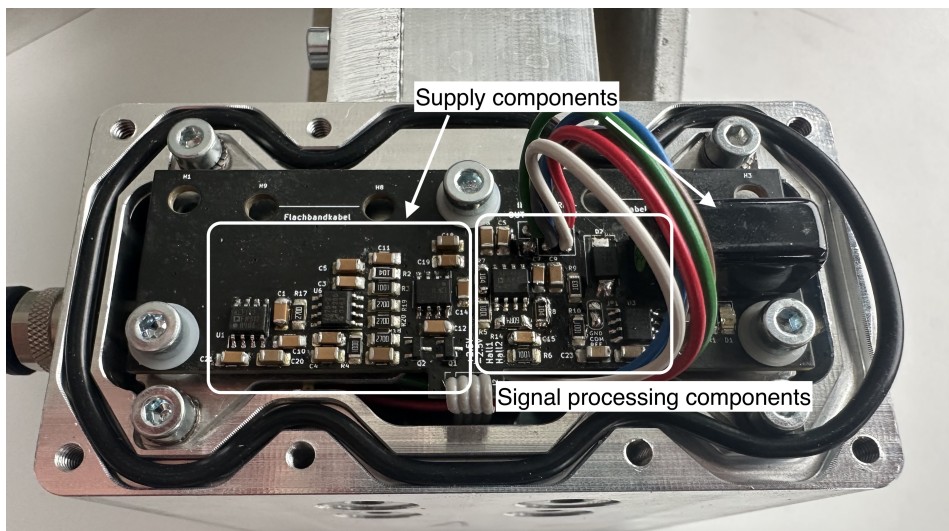

**Figure 10.** Populated and installed mainboard mounted inside the detector. The closed cables are connected to the outgoing port for supply and input and output signals. The grey flatland cable connects the sensors to the mainboard.

### 3. Sensor Characterization

For a reliable detection of the fastening, the sensors have to detect minute changes in the magnetic flux density caused by the fastening's presence. Hence, they have to feature good noise performance and a well known temperature profile for continued operation periods. Therefore, a detailed sensor characterization had to be conducted especially for the Honeywell sensor due to its limited data sheet. For this purpose a characterization circuit was designed, consisting of a REF01CSZ (for the Honeywell sensor) or a REF192ESZ (for the SiLabs sensor) voltage reference IC and the Hall effect sensor. The characterization circuit was placed inside a so-called $\mu$-metal box. This box with highly permeable double layered metal walls shields the electronics from external magnetic fields and so allows unhindered measurements. For the temperature drift measurements, the box was placed inside a climate chamber to perform controlled temperature tests. The measurement equipment consisted of a digital multimeter (DMM) of type HP 34401a, which was controlled over GPIB by MATLAB R2020a [15] on a laboratory computer. The measurement set-up is displayed in Figure 11.

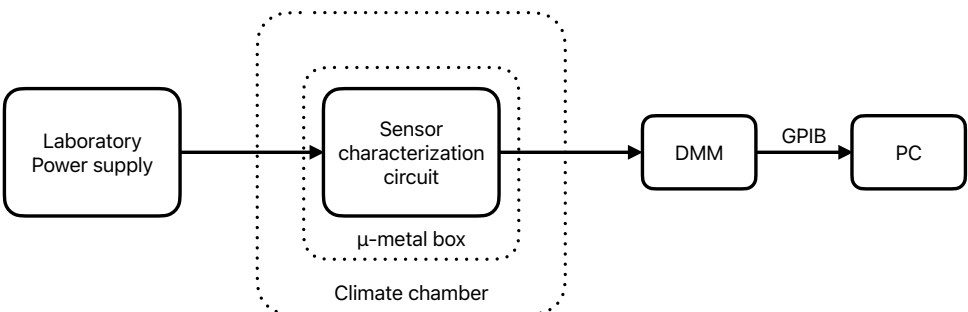

**Figure 11.** Measurement set-up for sensor characterization. The characterization circuit with the sensor under test is placed inside the $\mu$-metal box which is placed inside the climate chamber if temperature drift measurements are conducted. The electronics are supplied by a laboratory power supply, while the output signals are measured using a digital multimeter. The measurements are stored on a PC.

To determine the standard deviation of the sensors's output signal versus time, the sensors's output signal was measured for 4.5 h with a sampling frequency of 1 Hz, during which the assembly was held at $T = 25\,°C \pm 0.5\,°C$ in the climate chamber. Without a noticeable time drift, the standard deviation $\sigma_{\hat{\mu}}$ of the mean $\hat{\mu}$ was calculated to be

$$\sigma_{\hat{\mu}} = 740\,\mu V$$

for the Honeywell sensor. The standard deviation of this measurement is smaller than the sensor's signal variation due to a third of the earth's magnetic field, which was our goal to achieve. Therefore, the sensor's noise characteristics will not hinder proper detection. The SiLabs sensor yielded results with a standard deviation of

$$\sigma_{\hat{\mu}} = 36.437\,mV\,,$$

which is way beyond acceptable levels.

Furthermore, the temperature drift of the Honeywell sensor was measured by again placing the characterization circuit inside the $\mu$-metal box and now the box inside a climate chamber. The initial temperature of the climate chamber was set to $-35\,°C$ and steadily increased to $80\,°C$ while the zero deflection output voltage was measured again with the DMM. As seen in Figure 12, the temperature drift of the sensor's output voltage can be modeled by an affine function with a slope of $-847\,\mu V\,°C^{-1}$.

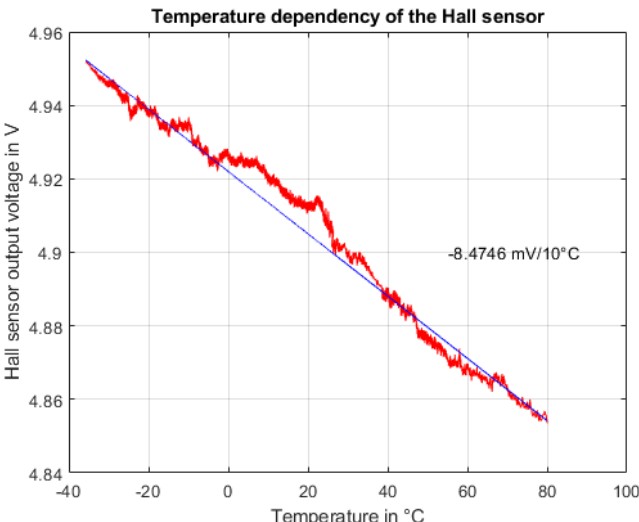

**Figure 12.** Temperature drift of the Honeywell SS94A1F Hall effect sensor. The drift can be modeled by a line with a slope of $-847\,\mu\text{V}\,°\text{C}^{-1}$.

The temperature drift of the SiLabs sensor is well defined by the data sheet to be less than $\pm 0.05\,\%\,°\text{C}^{-1}$ in a range of $0\,°\text{C}$ to $70\,°\text{C}$ which relates to $\pm 12.5\,\text{mV}\,°\text{C}^{-1}$ based on $2.5\,\text{V}$ zero field output. Therefore, an extensive temperature drift measurement was not conducted. Due to the common mode rejection of the instrumentation amplifier, the temperature drift will be cancelled out for similar sensors exposed to identical temperature environments. When the sensors are used in other configurations or for different purposes, however, the temperature drift can play a significant role. Since the detector is still in its prototyping phase, the signal processing and placing and combination of the sensors can still be part of future modifications in which the temperature drifts of two sensors might not be canceled out.

The Honeywell sensor's noise characteristics were measured by a spectrum analyzer in the frequency range of interest. The frequency range of interest of $500\,\text{Hz}$ is specified by the tamping machine's maximal velocity of $5\,\text{km}\,\text{h}^{-1}$ during operation and the required spatial resolution for the robust detection of fasteners (od. sleepers) of $20\,\text{mm}$ for the given maximum velocity of $5\,\text{km}\,\text{h}^{-1}$. This results in a minimal detection frequency of $69.5\,\text{Hz}$. Figure 13 depicts the spectral noise performance of the Honeywell sensor. Over the appointed frequency range of $500\,\text{Hz}$, the noise voltage $V_{\text{noise}}$, with a reference resistance of $50\,\Omega$, can be calculated to be $0.594\,\text{mV}$.

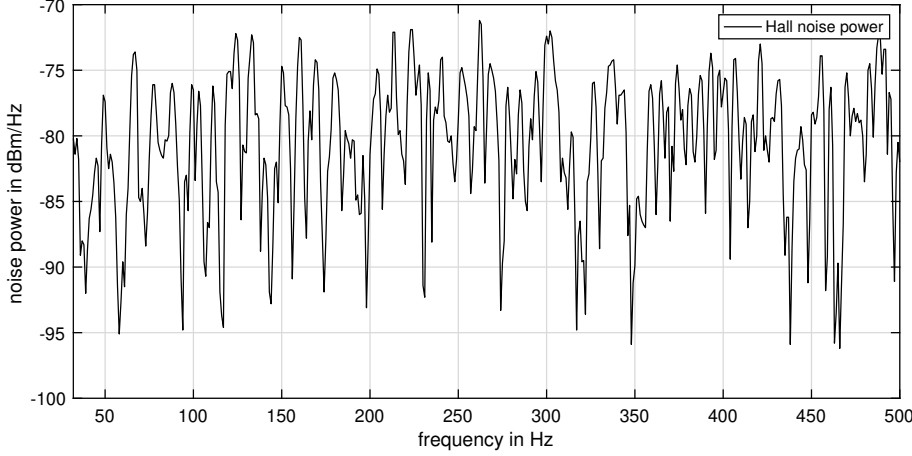

**Figure 13.** Noise spectral density of the Honeywell SS94A1F Hall effect sensor over a frequency range of $500\,\text{Hz}$.

The noise power of the SiLabs sensor is given in the data sheet as $1\,\mu V/\sqrt{Hz}$. By multiplying with the square root of the frequency range of interest of 500 Hz, the noise voltage $V_{\text{noise}}$ results in 22.36 µV. Due to the detailed data sheet, further noise measurements were not conducted with the SiLabs sensor. So, both sensors exhibit only minor noise levels and so fit well for our purpose. All inspected sensor characteristics are summarized in Table 2.

**Table 2.** Inspected characteristics of the used magnetic sensors.

| | SS94A1F | Si7211 |
|---|---|---|
| $\sigma_{\hat{\mu}}$ | 740 µV | 36.437 mV |
| temp.-coeff. | $-847\,\mu V\,^{\circ}C^{-1}$ | $\pm12.5\,mV\,^{\circ}C^{-1}$ |
| $V_{noise}$ | 0.594 mV | 22.36 µV |

## 4. Experiments and Results

For validation of the developed detector system, two configurations were analyzed each in a laboratory and a real environment. The first configuration with two `REF192` voltage reference ICs to supply the SiLabs sensors was designed with an extra sensor board at the bottom of the air gap facing side of the yoke. The second configuration features two `REF01` voltage reference ICs to supply the Honeywell sensors. These sensors were directly mounted onto the down facing side of the yoke without a separate sensor board. Both set-ups were first tested in a laboratory set-up (Figure 14), with results shown in Figure 15, and then tested on real railway tracks with steel sleepers on a measurement trolley (Figure 16).

### 4.1. Laboratory Set-Up

Figure 15 displays the measurement results of the laboratory set-up. The SNR of the SiLabs sensor at the top of the Figure is low, compared to the Honeywell sensor at the bottom. For that reason, a simple threshold detector to solve the detection problem at hand seems inadequate for this setup. On the bottom of the subplot, the Honeywell sensor features acceptable SNR and a simple threshold detector could potentially be implemented. Again, especially side-by-side, it can be observed, that the Honeywell sensor yields clearer results in laboratory set-ups. For further test measurements on real railway tracks, similar results were thus expected.

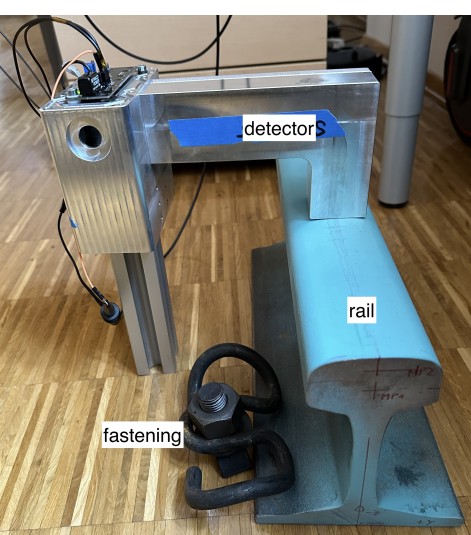

**Figure 14.** Laboratory set-up to test both configurations before moving to real railway tracks. The detector is supported by a piece of aluminum. The fastening is moved in rail direction underneath the detector.

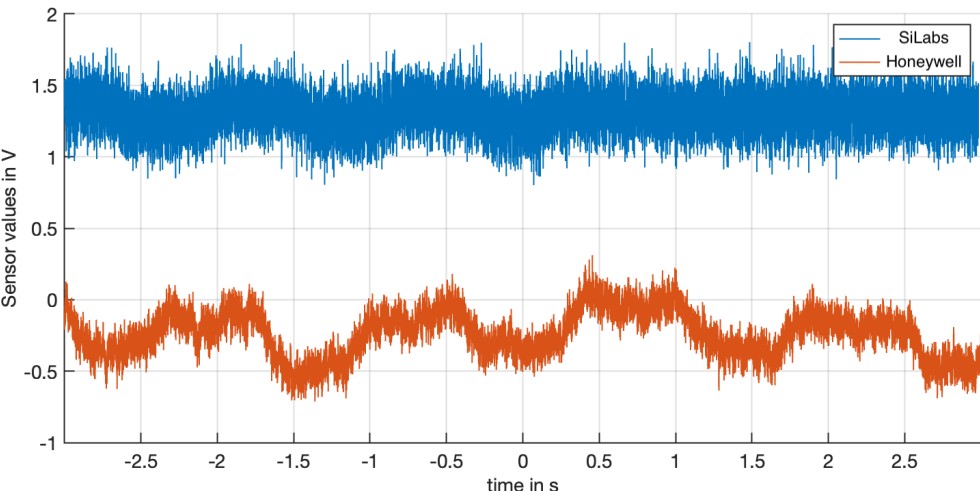

**Figure 15.** Test measurements with the SiLabs sensor at the top and the Honeywell sensor at the bottom in a laboratory set-up. Due to higher needed gain settings the signal-to-noise ratio of the SiLabs sensor's reading is clearly lower than the reading of the Honeywell sensor.

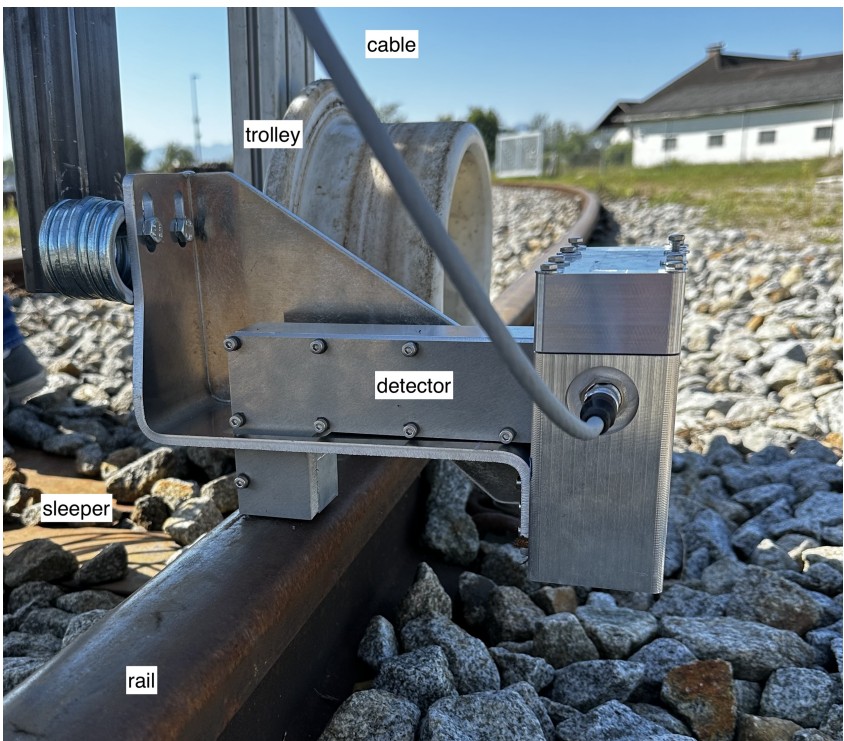

**Figure 16.** Detector mounted on the measurement trolley, connected to the Picoscope oscilloscope and a power supply outside of the picture frame by a shielded cable.

### 4.2. Real World Validation

Real world tests on railway tracks with steel sleepers were conducted using a measurement trolley, a vehicle to move mounted sensor prototypes on tracks with speeds comparable to machine speeds during operation. The trolley was manually moved over the tracks passing several sleepers during measurements. The instrumentation amplifier output was monitored by a PicoScope 3403D [16] oscilloscope with a vertical scale of 2 V/div and a sampling frequency of 10 kS/s. In the final application, as part of the track maintenance machinery, a PLD will be used for digitization and digital signal processing. A laboratory power supply supplied the detector with 24 V. As can be seen in Figure 16, the rail facing yoke was mounted as close to the rail as possible without touching it. The air

gap facing side has its lowest point right above the rail tracks top face height. Furthermore, the sensor case is designed to be mounted to a tamping machine to withstand both wet and dusty conditions during on-track maintenance operations. Several bolts are used to securely hold all parts together and to avoid any gaps and so ensuring water- and dust tightness. Additionally, all components are made from non-magnetic materials like aluminum and stainless steel.

The measurement results of the SiLabs sensor configuration are shown in Figure 17 in the top subplot (a). Again, the SNR is low and a threshold detection with unprocessed signals seems impossible. Due to simpler fabrication and usage of SMD parts, some signal processing was considered to counteract low SNR. Therefore, a digital low-pass filter with filter order $N = 50$ and a cut-off frequency $f_c = 70\,\mathrm{Hz}$ was implemented and tested on measurement signals in subplots (a) and (c) of Figure 17. As shown in subplots (b) and (d) of Figure 17, the filter yields higher SNR and a higher detection probability for both the SiLabs sensor and the Honeywell sensor configurations. The seemingly arbitrary signal base line changes due to inconsistent measurement trolley movement. If the trolley were pressed to the tracks as real tamping units are, the baseline is assumed to be stable due to a constant air gap between the yoke and the rail. Furthermore, vibrations will occur due to movement of the service machine and the operating tampering unit. The chosen differential sensing approach utilizing two magnetic field sensors, in addition to signal filtering and averaging, significantly reduces such influences from external vibrations, enabling robust sleeper detection in real-world operation scenarios. Additionally, the local maxima are quite broad compared to the plotted fastenings due to the sleepers material, steel. For standard steel concrete sleepers, mainly the fastening would cause a sensor deflection. Here, both the fastening and the steel sleepers act on the magnetic field profile.

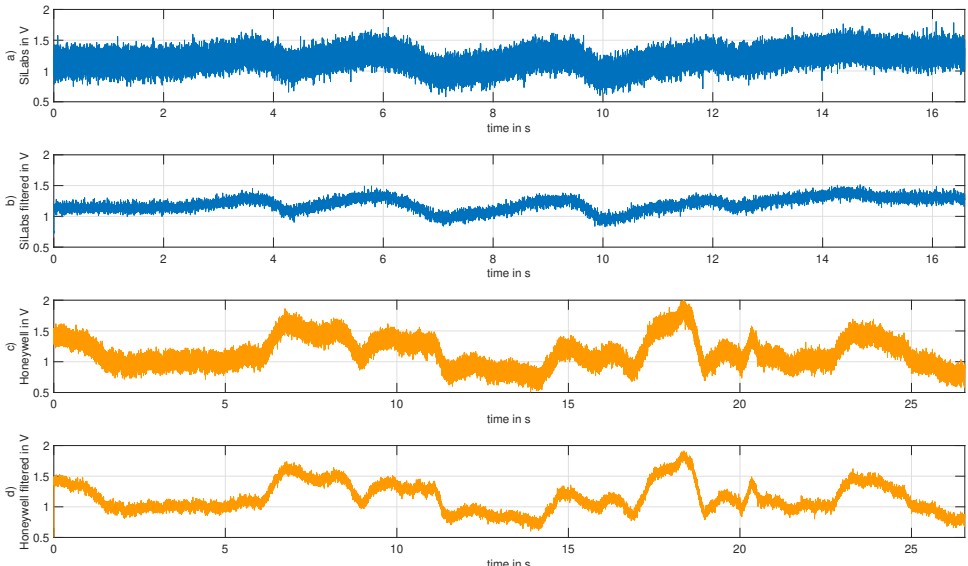

**Figure 17.** Measurements using the SiLabs sensor configuration in the laboratory set-up yields unusable results as seen in subplot (**a**). Computing a digital FIR low-pass filter with filter order 50 and a cut-off frequency of 70 Hz yields usable results for the SiLabs sensor shown in subplot (**b**). The Honeywell sensor signal, seen in subplot (**c**) in orange, delivers the best SNR and promises highest detection probability. Subplot (**d**) shows the Honeywell sensor signal filtered with the before mentioned settings. A value of 0.75 V was added to the SiLabs sensor signals for better comparison between the different sensors's results.

Figure 18 now shows an example implementation of a simple slope detector. Due to the signals behavior around the fastenings where the signal either steeply falls or rises, the slope detector analyses the value difference of a current sample with samples around and determines if the slope is great enough for a fastening to be detected. Additionally, due to a

certain slope behavior before the fastenings, the slope detector checks for a certain slope before the current sample to improve detection probability. A pseudo code is given below:

```
for i = α to length(signal)
    if abs(signal(i) - signal(i+k)) > threshold1 AND ...
                            ...abs(signal(i) - signal(i-α)) < threshold2
        detected(i+k/2) = 1
```

In subplot (a) the Honeywell sensor signal is shown in mT. The output of the instrumentation amplifier was back converted using the amplifier's gain and the sensors's sensitivity. It can be seen, that the magnetic flux density variation due to the fastenings is of marginal size. Additionally, the real fastening positions are marked in red. It can be observed, that the fastening is located between local maxima of the sensor signal, where the signal shows high slope. These slopes result from the differential measurement approach. Ideally, when the detector is located exactly over the fastening, both sensors should be penetrated by an equally strong magnetic field and therefore output the same value.Due to the differential measurement approach, the output of the instrumentation amplifier in this case approaches zero and negates its output, which marks the fastening position directly at the zero-crossing. The resulting signal slightly resembles one period of a harmonic signal. In real world scenarios like this, the sensors never output exactly the same value and the offset is never zero. Therefore, their outputs never cancel out. Hence, the different slopes and offsets in the signal. Besides, at $t \approx 19\,\mathrm{s}$ two dips can be observed. At this point there were two sleepers directly next to each other. Additionally, the first sleeper was not detected, since the trolley speed, as accelerated by hand, was too slow and the slope therefore rising too weakly. The sleeper at $t \approx 16\,\mathrm{s}$ was not detected due to abnormal signal variation at that point possibly as a cause of unintentional lateral trolley movement. Thus, Figure 18 shows, that, applying a median filter to remove outliers, the magnetic sleeper/fastening detection is possible even with this rough draft of a prototype. With further processing in future works, the slopes can be detected to precisely locate the fastenings and so the sleepers.

Additionally, since the measurement trolley was moved by hand and thus not with constant speed, the sleepers are detected with a different spacing between each other. If sensor signal versus trolley location was plotted, the sleepers would be equally spaced. Due to a sampling frequency of $1000\,\mathrm{Hz}$, the maximum theoretically possible detection resolution is given by $1.39\,\mathrm{mm}$ for an assumed maximum trolley velocity of $5\,\mathrm{km\,h^{-1}}$.

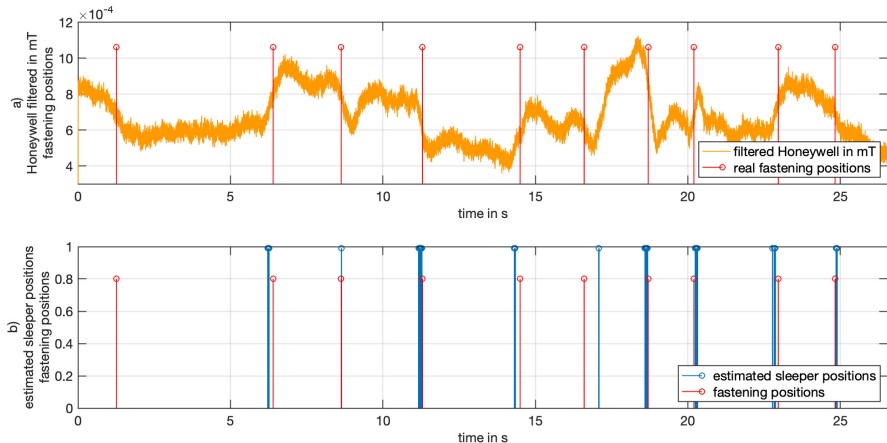

**Figure 18.** Subplot (**a**) shows the filtered Honeywell sensor signal as calculated as magnetic flux density and given in mT. It can be seen, that due to the differential measurement, the fastening is located at the parts of high slope of the signal. In subplot (**b**) a slope detector is displayed in blue. In both subplots the actual fastening positions are displayed in red.

*4.3. Discussion of Results*

Comparing both sensor configurations, it can clearly be seen that the Honeywell sensor has a signal-to-noise ratio advantage over the SiLabs sensor. Even though the mounting of the Honeywell through-hole sensor is more complicated, detection performance, as seen in subplot d) of Figure 17, outweighs simplicity in this case. As expected, the developed electronics are sensitive enough to handle even minute changes in signal variation. Additionally, applying a low-pass filter on the sensor output signals improves detection performance. Therefore, the development of a detector with the Honeywell sensors will be pursued. Furthermore, it could be shown, that magnetic sleeper detection is possible even under adverse conditions in real-world scenarios.

## 5. Conclusions and Outlook

Throughout this paper we describe the concept, design, implementation and validation of a novel sensor system capable of detecting sleepers in railway systems even when covered or buried in gravel. Simulation studies have been utilized to optimize the magnetic circuit and sensor geometry under the given application constraints, in particular for the required large air gap of >150 mm between the sleepers and the detector structure. Low noise sensor electronics, developed for two target Hall sensor prototypes, provide a decent sensor performance capable of detection of sleepers in laboratory, as well as realistic track situations.

Future work will focus on improvements of the analog electronic circuitry for optimized pre-filtering of the analog Hall sensor signals in the analog processing chain, and the following digital processing of the digitized detector signals within the application PLD system for robust and reliable detection of sleepers in all relevant practical track maintenance situations. Additionally, the detector will be mounted on tampering machines in the near future to further validate its performance and reliability under real-world operation conditions. With more data the detection algorithm will be further adapted and enhanced.

## 6. Patents

As a major result of this research, the detector and its functionality have been patented in patent [17], H. Hüttmayr, WO 2023/081946 A1, European Patent, "Device for detecting crossties of a track", 19 May 2023.

**Author Contributions:** Conceptualization, L.H. and B.Z.; methodology, L.H. and B.Z.; software, L.H.; validation, L.H., B.Z. and T.T.; formal analysis, L.H. and B.Z.; investigation, L.H. and B.Z.; resources, L.H. and B.Z. and H.H.; data curation, L.H.; writing—original draft preparation, L.H. and B.Z.; writing—review and editing, L.H., B.Z. and T.T.; visualization, L.H.; supervision, B.Z. and T.T.; project administration, B.Z.; funding acquisition, B.Z. All authors have read and agreed to the published version of the manuscript.

**Funding:** This research was funded by the Austrian Research Promotion Agency (FFG) and by the LCM K2-Center' Symbiotic Mechatronics' within the framework of the Austrian COMET–K2 Programme.

**Data Availability Statement:** The resulting data can be made available upon request.

**Acknowledgments:** The authors greatly acknowledge the Institute of Electrical Drives and Power Electronics, JKU Linz, for their support with our simulations.

**Conflicts of Interest:** Author Harald Huttmayr was employed by the company System7 Rail Support GmbH. The remaining authors declare that the research was conducted in the absence of any commercial or financial relationships that could be construed as a potential conflict of interest.

## Abbreviations

The following abbreviations are used in this manuscript:

| | |
|---|---|
| LOHET | Linear Output Hall Effect Transducer |
| SNR | Signal-to-Noise Ratio |
| SMD | Surface Mount Device |
| THT | Through-hole Technology |
| IC | Integrated Circuit |
| FEM | Finite Element Method |
| LiDAR | Light Detection and Ranging |
| PLD | Programmable Logic Device |
| DMM | Digital Multi Meter |
| GPIB | General Purpose Interface Bus |

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
