# Peer review of "Magnetic Railway Sleeper Detector"

_electronics, doi:10.3390/electronics13204005_

Round 1
Reviewer 1 Report
Comments and Suggestions for Authors
1) title is too short so that readers could not understand the paper's contribution;
2) Have you try pulsed eddy current testing? such as Pulsed eddy current technique for defect detection in aircraft riveted structures, NDT & E International 43 (2), 176-181
3) have you compared with other methods?
4) experimental results should be improved using some new signal processing methods.
thanks
Author Response
Comment 1: title is too short so that readers could not understand the paper's contribution;
Response 1: From our point of view the title is spot on describing what the paper is about.
Comment 2: Have you try pulsed eddy current testing? such as Pulsed eddy current technique for defect detection in aircraft riveted structures, NDT & E International 43 (2), 176-181
Response 2: We had a long thought process on how we could implement the detector and also thought about AC excitation. The problem here, as you can make out from the measurement results is that the signal changes are extremely low due to large air gap. DC approaches made most sense for us due to the high field properties of permanent magnets.
Comment 3: have you compared with other methods?
Response 3: As mentioned above, we thought about many possible detection methods, including capacitive sensing, AC magnetic field sensing, GPR,...
Comment 4: experimental results should be improved using some new signal processing methods.
Response: Exactly. Therefore, we now mention in the paper, that the detector will be used for the first time on real tampering machines in the following months this year. However, the results are too far in the future for this paper.
Reviewer 2 Report
Comments and Suggestions for Authors
This paper introduces an innovative method for detecting railway sleepers using magnetic anomaly signals, capable of identifying sleepers that are not visible to the naked eye. While the method shows promise, several questions and suggestions are raised regarding the content of the study:
1、Simulation Model Parameters: What are the key dimensional and material property parameters for each component in the simulation model? Presenting this information in a tabular format would greatly enhance the clarity and understanding for readers.
2、Impact of Vibrations and Magnetic Field Fluctuations: In practical applications, the vibration of the detection module and external magnetic field fluctuations are often inevitable. How might these factors influence the accuracy of detection? What strategies can be employed to minimize or mitigate their effects?
3、Consistency of Data in Figures 15 and 17: Do Figures 15 and 17 represent the same set of experimental data? Were these data obtained under controlled laboratory conditions, or from real-world field tests? Additionally, why are the same data presented in both Sections 4.1 and 4.2?
4、Limited Experimental Data: The paper provides only one set of experimental results, which may be insufficient to fully demonstrate the reliability and robustness of the proposed detection method. It is recommended that the author expand the dataset to strengthen the validation of the approach.
5、Clarification on "Spatial Resolution": In line 388, the term "spatial resolution" is mentioned. Does this refer to the spatial resolution of the sensor, or the resolution of the data sampling points? Clarifying this distinction would improve the precision of the discussion.
Author Response
Comment 1: Simulation Model Parameters: What are the key dimensional and material property parameters for each component in the simulation model? Presenting this information in a tabular format would greatly enhance the clarity and understanding for readers.
Response 1: I added the key parameters of the materials used in the simulation as a table.
Comment 2: Impact of Vibrations and Magnetic Field Fluctuations: In practical applications, the vibration of the detection module and external magnetic field fluctuations are often inevitable. How might these factors influence the accuracy of detection? What strategies can be employed to minimize or mitigate their effects?
Response 2: Vibrations are going to occur for sure. We think, that by using two detector on each outer side of the rail will counteract vibration sensitivity in addition to averaging and filtering. Vibration signals will probably be in a higher frequency range than our measurement signal.
Comment 3: Consistency of Data in Figures 15 and 17: Do Figures 15 and 17 represent the same set of experimental data? Were these data obtained under controlled laboratory conditions, or from real-world field tests? Additionally, why are the same data presented in both Sections 4.1 and 4.2?
Response 3: We chose another signal for Figure 17 to be more obviously different. The data of Figure 15 was obtained under laboratory conditions as seen in Figure 14. Figure 17 shows data from real world measurements using the measurement trolley as seen in Figure 16. To your second question, I hope the data is now more clearly different.
Comment 4: Limited Experimental Data: The paper provides only one set of experimental results, which may be insufficient to fully demonstrate the reliability and robustness of the proposed detection method. It is recommended that the author expand the dataset to strengthen the validation of the approach.
Response 4: We agree. At this point we only measured with the measurement trolley. In the following months, still this year, the detector will be mounted on a real tampering machine. Then both for the SiLabs sensors and the Honeywell sensors new data will be obtained. Since these measurements are too far in the future for this paper, only the already included data can be shown.
Comment 5: Clarification on "Spatial Resolution": In line 388, the term "spatial resolution" is mentioned. Does this refer to the spatial resolution of the sensor, or the resolution of the data sampling points? Clarifying this distinction would improve the precision of the discussion.
Response 5: With spatial resolution the detection precision of the fastening is meant. How precise can the fastening be detected using the detector. I added soe explanation to the paper.
Reviewer 3 Report
Comments and Suggestions for Authors
1. Please explain the advantages of this work compared to commercially available variable reluctance sensors, which are more robust, simpler, and readily available.
2. The overall magnetic circuit shown in Figure 3 is a traditional design. Please clarify the novelty of this work.
3. Please explain the rationale for using three magnetic field sensors instead of two or five…
4. It appears that the design parameters are not clearly outlined in this manuscript, and the choice of arm length seems somewhat arbitrary. Please explain and clarify this point.
5. Please clarify the novelty of the "Electronic Circuit Design." Alternatively, you may cite a relevant reference and remove any redundant information.
6. Please explain the reason for choosing this Hall effect sensor (Honeywell SS94A1F). Additionally, outline the parameters to consider when selecting an appropriate sensor for this application.
7. Please explain the reason for characterizing the commercial Hall effect sensor (SS94A1F) in this manuscript. Typically, this information can be found in the relevant datasheet.
8. Kindly include the low-frequency noise measurement results that reflect the flicker noise effect.
9. Please clarify whether the authors are the first to perform magnetic railway sleeper detection. If not, kindly include a table for comparison with previous works.
Comments on the Quality of English LanguageThere are some minor English mistakes in the manuscript. For example, the sentence starting in line 40 is too long, and the "to" after "additionally" should be removed. Furthermore, there are instances of missing definite or indefinite articles.
Author Response
Comment 1: Please explain the advantages of this work compared to commercially available variable reluctance sensors, which are more robust, simpler, and readily available.
Response 1: The hard part of our detection process is the precise localization over the huge air gap. From our point of view variable reluctance sensors are not able to achieve similar results over this great distance.
Comment 2: The overall magnetic circuit shown in Figure 3 is a traditional design. Please clarify the novelty of this work.
Response 2: The novelty does not lie in the for of the detector. Rather the application and the optimization for the huge air gap are novel. There probably is a better geometry for fastening detection in our case, but simulations to find this geometry would greatly increase simulation cost and time and was therefore not conducted. I tried to add some explanation to the paper.
Comment 3: Please explain the rationale for using three magnetic field sensors instead of two or five…
Response 3: As we now added to the paper, three sensors give an overview of the rough field distribution at the yoke's down facing side. More sensors would increase resolution, but also simulation time.
Comment 4: It appears that the design parameters are not clearly outlined in this manuscript, and the choice of arm length seems somewhat arbitrary. Please explain and clarify this point.
Response 4: We added some explanation to the paper. The material properties are now added to the paper. The arm length was chosen to lie between "too short so that field lines would potentially already close before even getting to fastening" and the outer edge of the track's clearance profile.
Comment 5: Please clarify the novelty of the "Electronic Circuit Design." Alternatively, you may cite a relevant reference and remove any redundant information.
Response 5: Of course the circuit is not a new invention. Again, the application is more important than the actual electronic design. Therefore, we now changed the circuit diagram to a block diagram and only described it more roughly.
Comment 6: Please explain the reason for choosing this Hall effect sensor (Honeywell SS94A1F). Additionally, outline the parameters to consider when selecting an appropriate sensor for this application.
Response 6: We now tried to describe the process and parameters to consider in more detail.
Comment 7: Please explain the reason for characterizing the commercial Hall effect sensor (SS94A1F) in this manuscript. Typically, this information can be found in the relevant datasheet.
Response 7: The datasheet provided by Honeywell lacks information. I hope we now clarify that in more detail in the paper.
Comment 8: Kindly include the low-frequency noise measurement results that reflect the flicker noise effect.
Response 8: We can gladly upload the data alongside the paper.
Comment 9: Please clarify whether the authors are the first to perform magnetic railway sleeper detection. If not, kindly include a table for comparison with previous works.
Response 9: According to our partner System7, no other tampering machine producer uses magnetic sleeper detection. Additionally, no other scientific publication could be found by our team.
Round 2
Reviewer 2 Report
Comments and Suggestions for Authors
Thank you for your thoughtful responses and revisions based on the previous review comments. After carefully reviewing the revised manuscript, I believe that the authors have adequately addressed all the issues raised in the previous round of review. The changes made have further strengthened the scientific merit and practical value of the paper.
Author Response
Response: Thank you for your feedback. We are glad to improve our work based on high quality review!
Reviewer 3 Report
Comments and Suggestions for Authors
Comments 2 and 3 are not properly answered.
Comments on the Quality of English LanguageA moderate English revision is required for this manuscript. For example, in line 4 "Nowadays still, maintenance personnel has to manually locate the sleepers if they cannot be detected by computer vision systems or visually by the operator" should be changed to "Even today, maintenance personnel still have to manually locate the sleepers if they cannot be detected by computer vision systems or visually by the operator.".
Author Response
Comment 1: Comments 2 and 3 are not properly answered.
Response 1: We tried to add some additional information to further strengthen the point that the geometry is not novel, but the application. Additionally, the choice of the number of sensors has been explained in more detail. Also, what could be improved with additional sensors.
Comment 2: English revision
Response 2: We revised our grammar and corrected (unmarked) several phrases.
Thank you for your review and constructive criticism